

# Research groups: How big should they be?

Isabelle Cook, Sam Grange and Adam Eyre-Walker

School of Life Sciences, University of Sussex, Brighton, United Kingdom

## ABSTRACT

Understanding the relationship between scientific productivity and research group size is important for deciding how science should be funded. We have investigated the relationship between these variables in the life sciences in the United Kingdom using data from 398 principle investigators (PIs). We show that three measures of productivity, the number of publications, the impact factor of the journals in which papers are published and the number of citations, are all positively correlated to group size, although they all show a pattern of diminishing returns—doubling group size leads to less than a doubling in productivity. The relationships for the impact factor and the number of citations are extremely weak. Our analyses suggest that an increase in productivity will be achieved by funding more PIs with small research groups, unless the cost of employing post-docs and PhD students is less than 20% the cost of a PI. We also provide evidence that post-docs are more productive than PhD students both in terms of the number of papers they produce and where those papers are published.

## INTRODUCTION

How large should a research group be? Should resources be concentrated into a small number of research groups or should funding be more evenly distributed? This question has been investigated in a number of different countries at a variety of different levels of organisation. Most analyses of individual research groups, rather than departments or universities, have found that the number of research papers per group member, is either unrelated (*Cohen, 1981*; *Johnston, Grigg & Currie, 1995*; *Seglen & Aksnes, 2000*) or that it declines with group size (*Brandt & Schubert, 2013*; *Carayol & Matt, 2004*; *Diaz-Frances, Ruiz-Velasco & Jimenez, 1995*). Reports that there is an optimal group size (*Qurashi, 1984*; *Qurashi, 1993*; *Stankiewicz, 1979*) appear to have limited statistical support (see for example Cohen's (*Cohen, 1984*) criticism of *Qurashi, 1984*), as do reports that productivity increases exponentially with research group size (*Wallmark et al., 1973*) (see criticism by *Cohen, 1981*).

The question of research group size and the allied question of funding has been brought back into focus with a recent analysis of National Institute of Health (NIH) data. Jeremy Berg, a former director at the NIH, found that both the number of papers and the median impact factor (IF) of papers increased with NIH funding per lab until a maximum was attained at approximately $750,000 per year, after which both the number

Corresponding author
Adam Eyre-Walker,
a.c.eyre-walker@sussex.ac.uk

of publications and the median IF declined (see https://loop.nigms.nih.gov/2010/09/measuring-the-scientific-output-and-impact-of-nigms-grants/, reported by *Wadman, 2010*). This has led to a policy by which grants from well-funded labs are subject to additional review by the NIH (*Berg, 2012*). However, Berg presented no statistical evidence in support of the maximum. A truer reflection of the data might be that groups with low levels of funding are relatively productive and that funding explains little of the variance in publication rate or where those papers get published. A recent analysis of the Canadian funding of science has come to a very similar conclusion (*Fortin & Currie, 2013*). Various measures of productivity and impact, including the number of publications and the number of citations, are positively correlated to the level of funding, but the relationship is very weak, both in terms of the slope of the relationship and the variance explained. Furthermore, the relationship is one of diminishing returns; productivity increases with funding but not proportionally (*Fortin & Currie, 2013*).

Here we consider the current relationship between research group size and productivity in the Biological Sciences in the United Kingdom. We consider several measures of productivity: the number of papers published by a research group, the impact factor of the journals in which those papers are published and the number of citations they receive as a function of group size.

# MATERIALS AND METHODS

## Group size

We emailed all principal investigators (PIs) in biological science departments in universities that had made a return to sub-panel 14 (Biological Sciences) of the 2008 edition of the Research Assessment Exercise. Email addresses were harvested from departmental web-sites. Emails were sent in two phases, in October 2012 and in October 2013 by IC and SG respectively. Contactees were asked to provide the number of post-docs, PhD students, technicians and other staff working in their group. If individuals were part-time or shared between faculty we asked that they be counted as a fraction of a full time equivalent. Contactees were also asked whether they had been at the same institution over the preceding 5 years, and only individuals that fullfilled this criterion were included in subsequent analysis; this was to make it possible to identify the publications produced by each PI. Copies of the emails sent to PIs are included as Supplemental Information 2.

## Publication data

The publications published by a PI were obtained by searching the ISI Web of Science database by employing their author search facility using last name, first initial and institutional address, restricting the search to papers published in the life sciences. To check the data, we listed the initials or first names associated with each paper returned by the initial search—for example a search of Jones, C at Dundee might return publications from Jones, Chris and Jones, Cate at Dundee. Publication files containing multiple authors were manually curated. We also checked all publication files that had papers with more than 20 authors since the address field in the initial search is not directly associated with

the author—for example, a search of Jones, C at Dundee might return a paper by Jones, C at Cambridge, who co-authored a paper with someone at Dundee. Such mistakes are more likely for papers with many authors. The publication data was downloaded by AEW in July and August 2014; however, only papers published between and including 2008 to 2012 were considered for scientists contacted in 2012, and between and including 2009 to 2013 for scientists contacted in 2013. For each publication, we divided the number of citations by the number of years since publication (e.g., the citations published for a paper in 2012 for individuals censused in 2012 would be divided by one, those from 2011 by 2. . . etc.). We also counted the number of authors for each paper and obtained the impact factor of the journal in was published using impact factors from 2013.

## Statistics

Because all the variables in this analysis have skewed distributions, we log transformed the data before performing normal least squares regression. However, we also performed regressions on untransformed data to gain further insight into the relationship between the number of papers and group size. For this we assumed the dependent variable was Poisson distributed and we estimated the parameters of the regression model by maximum likelihood; note that this differs from standard Poisson regression because we are assuming that the untransformed dependent variable is a linear function of the dependent variables, whereas regular Poisson regression assumes the log of the dependent variable is a linear function. Differences between nested models were tested using likelihood ratio tests and differences between un-nested models were assessed using the Akaike Information Criterion (AIC).

To test whether the regression coefficients (i.e., the slopes associated with PhDs, post-docs, technicians and other scientists) were significantly different to each other in a multiple regression, we bootstrapped the data by PI, re-running the multiple regression calculating the difference between the regression coefficients each time. We repeated this 1,000 times. The *p*-value was the proportion of differences between regression coefficients that were greater or less than zero, as appropriate (e.g., in testing whether the number of papers is more strongly dependent upon the number of post-docs and PhD students, where the regression coefficient for post-docs is greater than that for PhD students, the *p*-value was the number of differences between the bootstrap coefficients that were negative).

## Data availability

The anonymised dataset is available in Table S1.

## Ethical considerations

It was not considered necessary to submit this study for ethical review given the nature of the project—simply requesting research group size information directly from PIs. All participants gave their written consent in the form of an email reply. All data was treated as confidential.

**Table 1 Correlations between group members.** The correlations between the numbers of PhD students, post-docs, technicians and other group members, and the levels of significance from a test of whether the correlation is significantly different from zero.

|  | Post-doc | Technician | Other |
|---|---|---|---|
| PhD | 0.27*** | 0.18*** | 0.017 |
| Post-doc |  | 0.21*** | 0.12* |
| Technician |  |  | 0.24*** |

Notes.
* $p < 0.05$.
*** $p < 0.001$.

## RESULTS

In order to investigate the relationship between scientific productivity and research group size within the biological sciences in the United Kingdom, we contacted all principal investigators (PI) who were in UK research departments that took part in the 2008 Research Assessment Exercise. In total, 2,849 academics were contacted personally by email, of which 398 (10%) replied and had been at the same institution over the previous 5 years. We required them to have been present at the same institution so that we could obtain their publication record over that period (see Materials and Methods). We asked them how many PhD students, post-docs, technicians and other researchers (mostly research associates) they had in their research group on the day they were contacted (hereafter we define a group as the principal investigator (PI) and their associated post-docs, PhD students, technicians and other research staff (usually pre-doctoral research assistants)). We subsequently downloaded their publications from the previous 5 years from the Web of Science along with the number of citations each paper had received over a period of 6 or 7 years, depending on when the PI was contacted (downloading of the Web of Science data was done sometime after the PIs were contacted). The number of citations for each paper was divided by the number of years since publication.

### Group size

Most biology research groups in the UK are of modest size, containing less than 10 staff and students, including the PI (Fig. 1). The mean research group size is 7.3 (standard deviation of 4.5) with a range of 1 to 31. On average a research group contains 3.0 PhD students, 2.1 postdocs, 0.5 techinicians and 0.68 other staff (mostly research associates). The numbers of post-docs, PhD students, technicians and other staff are mildly but significantly positively correlated to each other, with the exception of PhD students and other staff (Table 1).

### Models

Since most of the variables in our analysis have strongly skewed distributions we log transformed the data, adding one where the variable contained zeros (this only applied to the number of publications, and the numbers of PhD students, post-docs, technicians and other researchers). This yielded approximately normally distributed variables. As a consequence, in fitting a linear model to the log transformed data we are fitting a model of

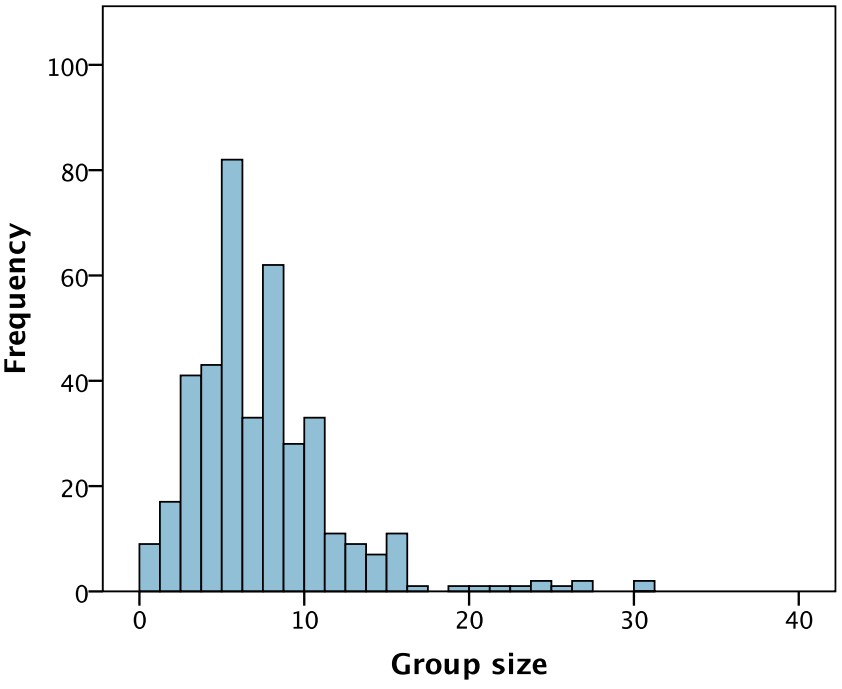

**Figure 1** **The distribution of group size amongst 398 PIs within the Life Sciences in the United Kingdom.**

the following form:

$$y = ax^b \qquad (1)$$

since

$$\log(y) = \log(a) + b\log(x). \qquad (2)$$

The slope of the relationship between $\log(y)$ and $\log(x)$ indicates whether $y$ increases proportionally with $x$ ($b = 1$), less than proportionally with $x$ ($b < 1$) (i.e., diminishing returns, or decreasing returns to scale), or more than proportionally with $x$ ($b > 1$) (i.e., increasing returns or increasing returns to scale). Unless otherwise clearly stated, references to a variable will mean the log of that variable—e.g., the number of papers is correlated to group size means the log of the number of papers is correlated to the log of group size.

### Number of publications versus group size

The average number of publications published by each group in the previous five years was 22.0 papers (SD = 18.8) but varies considerably between PIs, from 0 to 177. The number of publications over the preceding 5 years is significantly correlated to the total group size ($r = 0.43$, $p < 0.001$) (Fig. 2). However, group size explains less than 20% of the variance in the number of papers, and at all levels of group size there is substantial variance in the number of papers produced (Fig. 2). The slope of the relationship between the number of papers and group size is significantly less than one ($b = 0.57$ (SE = 0.06),
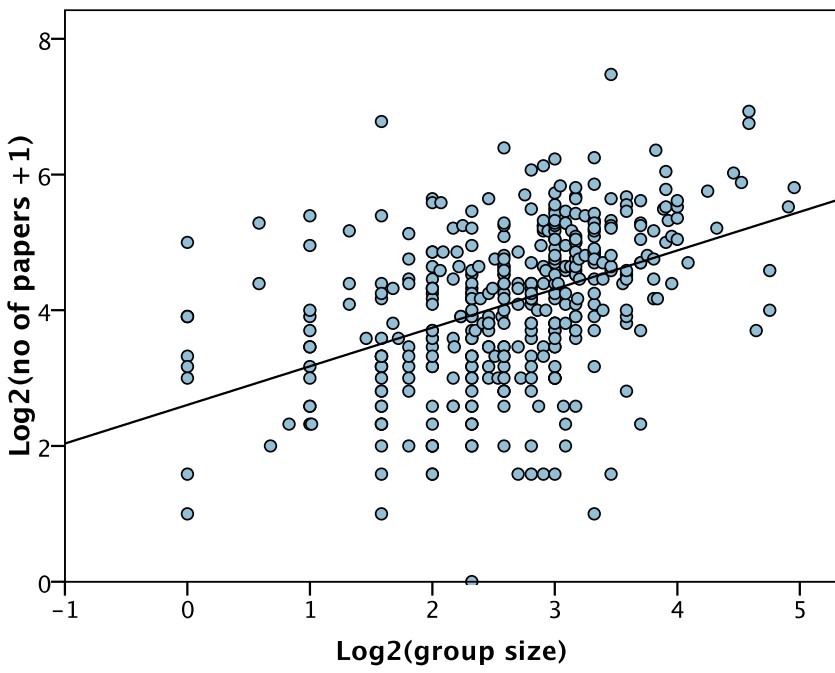

**Figure 2  Number of papers versus group size.** The least squares line of best fit is shown.

$p < 0.001$) indicating a diminishing returns relationship; i.e., the number of papers increases with group size but less than proportionally. This is not simply due to adding one to the number of papers before log transforming, because if we do not add one and drop the one group with no research papers, we get qualitatively the same relationship: $b = 0.62$, $p < 0.001$). The diminishing returns relationship can be illustrated simply by dividing the (untransformed) number of papers by the (untransformed) group size; the number of papers per group member decreases as group size increases (Spearman's rank correlation $= -0.20$, $p < 0.001$) (Fig. 3).

A multiple regression suggests that the number of published papers is significantly and positively correlated to the number of PhDs ($p < 0.001$), post-docs ($p < 0.001$) and other researchers ($p = 0.001$), but not the number of technicians ($p = 0.47$). The slope associated with post-docs ($b = 0.39$) is considerably larger than the slope associated with either PhD students ($b = 0.24$) or other researchers ($b = 0.22$). In the case of post-docs versus other members this difference is significant ($p = 0.032$) and almost significant for PhDs versus post-docs ($p = 0.058$). The slopes suggest that post-docs are on average more productive than PhD students or other researchers.

In the biological sciences most papers are co-authored, often with a large number of co-authors—the mean number of authors per paper considered here is 9.3. As a consequence the number of papers may not reflect the output of a particular research group but the collaborations the group participates in. We therefore also considered the number of papers in which the PI was first or last author (these are traditionally the places where the lead PI on a project will appear in the biological sciences). On average, each PI produced 11.6 (SD $= 10.0$) first and last author papers, which means that about half of

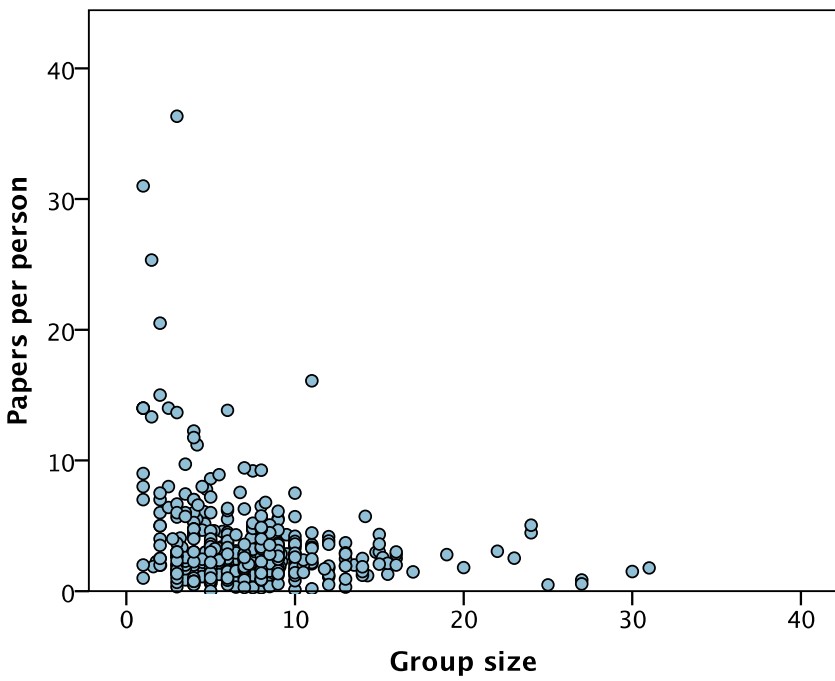

**Figure 3 Paers per group member versus group size.** The number of publications per group member versus group size.

all papers associated with a PI are first and last author papers. However, the proportion of papers that are first author papers varies significantly between PIs (Chi-square $= 1{,}455$, df $= 378$, $p < 0.001$). The proportion is not surprisingly significantly negatively correlated to the number of authors on a paper ($r = -0.18$, $p < 0.001$), but it is not correlated to group size ($r = -0.019$, $p = 0.70$).

The number of first and last author papers is significantly correlated to group size ($r = 0.39$, $p < 0.001$), with a slope ($0.56$, $p < 0.001$) which is similar to that observed for the total number of publications, indicating diminishing returns with increasing group size. The number of first and last author papers is significantly corelated to the number of PhD students ($b = 0.33$, $p < 0.001$) and post-docs ($b = 0.41$, $p < 0.001$), but not other members and the number of technicians. The slopes for PhDs and post-docs are not significantly different to each other in this analysis ($p = 0.24$).

## Impact versus group size

Large research groups produce more papers than small groups, but do they also produce papers that appear in journals with higher IFs and which gain more citations? We find that the mean IF of the papers associated with a PI is significantly correlated to group size ($r = 0.14$, $p = 0.004$) (Fig. 4). However, the correlation is very weak—group size only explains 2% of the variance—and the slope is very shallow ($b = 0.10$, $p = 0.004$). Surprisingly we find that the mean IF is positively correlated to the number of post-docs ($b = 0.22$, $p < 0.001$) and other researchers ($b = 0.087$, $p = 0.022$), but is negatively correlated to the number of PhD students ($b = -0.16$, $p < 0.001$). These slopes are

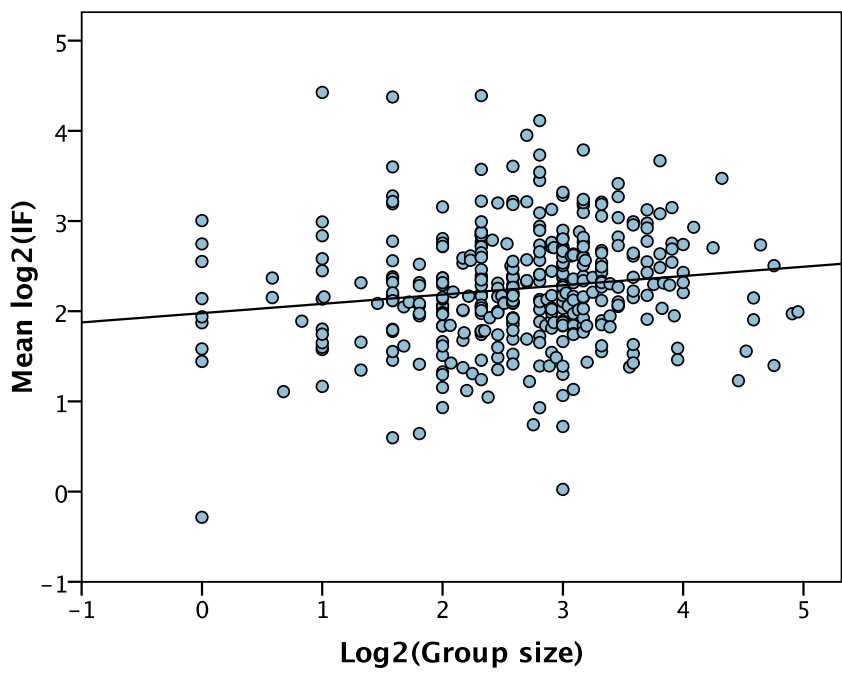

**Figure 4  IF versus group size.** The least squares line of best fit is shown.

significantly different to each other (PhDs versus post-docs, $p < 0.001$, PhDs versus others, $p < 0.001$, post-docs versus others $p = 0.02$).

Much the same pattern holds for the number of citations per year. The mean log number of citations per year is significantly correlated to group size ($r = 0.15$, $p = 0.004$) (Fig. 5). However, the regression explains very little of the variance and the slope is very shallow ($b = 0.12 \, (0.04)$). The number of citations is only significantly correlated to the number of post-docs ($b = 0.19$, $p < 0.001$).

## DISCUSSION

We have shown that the number of papers published by a group, and the mean IF and the mean number of citations of those papers, are all positively correlated to group size. However, the slopes are very shallow for the IF and the number of citations, indicating that group size has little effect on where papers are published and how many citations they receive. For all three variables, the slope of the relationship between the log of the variable and the log of group size is less than one, indicating that each variable increases with group size, but less than proportionately; i.e., there is diminishing returns such that doubling the group size leads to less than a doubling in measures of productivity.

A diminishing returns relationship could potentially be due to two non-mutually exclusive factors. It could be that all group members are equally productive, but that increasing group size leads to inefficiencies that lead to a reduction in the productivity per individual. Alternatively, it might be that the PI has a larger effect on productivity than other group members. Since there is generally only one PI per group, increasing group size leads to a dilution of this contribution and hence diminishing returns. Given

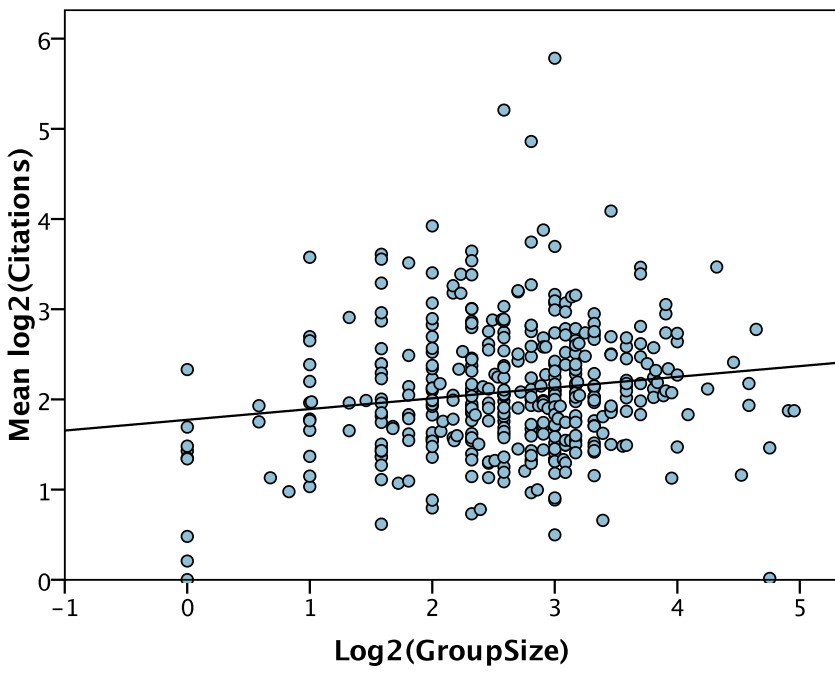

**Figure 5 Number of citations per year versus group size.** The least squares line of best fit is shown.

that post-docs appear to be significantly more productive than PhD students, it seems quite possible that PIs contribute more to productivity than post-docs, given their years of additional training, and that diminishing returns arises because of this.

To investigate the reasons for the diminishing returns relationship we fit a series of models to the untransformed data using maximum likelihood assuming the error term was Poisson distributed (this is different to Poisson regression in which the log of the dependent variable is assumed to be a linear function of the independent variable). We concentrated on the relatiomship between the number of papers and group size because the other variables are only very weakly correlated to group size. We subtracted the PI from the group size because groups cannot have zero members. First, we compared a simple linear model ($y = a + bx$) to a model including a quadratic term ($y = a + b_1x + b_2x^2$); the inclusion of the quadratic term did not significantly improve the fit of the model (log likelihood for the linear model $= -2927.89$, for quadratic model $= -2927.35$) suggesting that the number of publications increases linearly with group size. However, one might argue that the quadratic model simply fails to improve the fit because it is an inadequate model. We therefore fit a very different model of the form $a + be^{-cx}$; this equation allows $y$ to be linearly related to $x$ (when $c$ is small) or to show diminishing or increasing returns, all with with arbitrary intercepts. This model fits the data slightly, but not significantly, better than the simple linear model (log likelihood $= -2927.47$). Furthermore, the value of $c$ is very small so this model is essentially linear; the estimated relationship is $313.5 - 303.7e^{-0.0065}x$ which is approximately $313.5 - 303.7(1 - 0.0065x) = 9.8 + 2.0x$. This is very similar to the simple linear model, $y = 10.1 + 1.9x$, fitted above. These analyses therefore suggest that the diminishing returns does not arise because larger groups

become inefficient, because the evidence suggests that each additional member increases productivity by the same amount as other members. Instead diminishing returns seems to arise because the PI contributes significantly more to productivity than other team members, at least when group size is small. We cannot say anything about the contribution of the PI in larger groups because the contribution of the PI to productivity might be a function of group size; for example, the contribution of the PI might decline as group size increases.

What do the results imply for the funding of science? Although, we have found that all measures of productivity increase with group size, they do not do so proportionally; there is diminishing returns. This would seem to suggest that we should favour small groups. However, it may be that some types of science can only be conducted by large research groups. Furthermore, PIs are more expensive to employ than other team members, so it might pay to employ more post-docs and PhD students than PIs. For example, if the average cost of employing other group members is 20% that of a PI then there no evidence of a diminishing returns relationship between the (log) of the number of papers and (log) cost since the slope = 1.0. If the relative cost is 10% there is a pattern of increasing returns because the slope = 1.5. Although, we do not know the average cost of employing PIs, post-docs and PhD students, it seems unlikely that PIs cost more than 5-times as much as the average post-doc and PhD student, at least not in the UK. The data therefore suggest that the best policy in terms of scientific productivity is to invest in more PIs, rather than post-docs and PhD students. Such a policy would also go someway to addressing the poor career prospects amongst PhD students and post-docs within academia. In the UK, only 3.5% of science PhDs gain a permanent academic position and 17% go into non-university research, with about 79.5% ending up outside scientific research completely (*The Royal Society, 2010*). However, it should be appreciated that recruiting more PIs is likely to reduce the average quality of PIs.

Our results are consistent with several previous studies in which productivity has shown a diminishing returns relationship with research group size for scientists in Germany, France and Mexico (*Brandt & Schubert, 2013*; *Carayol & Matt, 2004*; *Diaz-Frances, Ruiz-Velasco & Jimenez, 1995*); a similar pattern is also evident between measures of productivity and research grant income (*Fortin & Currie, 2013*; *Wadman, 2010*) (note that Berg did not explicitly test for diminishing returns in his analysis of NIH data, but the linear regression had a significantly positive intercept which would yield a diminishing returns relationship on a log scale). In contrast, *Cohen (1981)* and *Seglen & Aksnes (2000)* reported a linear relationship between the untransformed number of papers and untransformed research group size with an intercept that was close to or not significantly different to zero. In the case of the analyses performed by Cohen this may have been due to power because for each of his three datasets he had rather little data.

We have also presented evidence that post-docs are more productive than PhD students. They produce more papers and those papers are published in journals with (slightly) higher IFs; in fact the mean IF significantly decreases as the number of PhD students increases, although the decrease is very slight. The number of citations is also only

significantly correlated to the number of post-docs but not PhDs. To obtain a more quantitative estimate of how much more productive post-docs were than PhD students in terms of the number of papers we re-ran the multiple regression on the untransformed data using the Poisson regression model. The slopes suggest that each post-doc adds 3.48 papers per 5 years, whereas PhD students and other researchers add 1.53 and 1.98 papers, respectively. The slopes are significantly different between PhDs and post-docs in the untransformed data ($p = 0.004$), but not between other researchers and either post-docs or PhDs. It is perhaps not surprising that post-docs are more productive than PhD students, since post-docs have more training and there is likely to be some degree of selectivity in which PhD students become post-docs.

Our definition of produtivity is limited. In particular, we do not take into account the role of PIs as teachers and the conribution that post-docs and PhD students make in careers other than academic research. Furthermore, we do not measure other forms of productivity such as patents and policy documents. Our results must be interpreted in the light of these caveats.

Although we have collected data from a large number of groups, we have relied upon self-reporting. This might have potentially biased the results. In particular, we may have had under-reporting from small groups or groups that were unproductive. It is difficult to address this problem. Site visits to selected universities may help, but even then there is no guarantee of complete or unbiased results. We have also restricted our analysis to PIs that have remained at the same instutition for 5 years; this might have biased our results away from young researchers, who may move early in their career.

In summary, we have shown that three measures of productivity, the number of papers, the impact factor and the number of citations, increase with group size. However, they all show a pattern of diminishing returns and the relationships are weak in terms of the variance that group size explains; this is particularly the case for the impact factor and the number of citations. Our analyses support a funding model in which productivity is maximised by having many small groups rather concentrating resources into a few large ones.

## ACKNOWLEDGEMENTS

We are grateful to Torben Schubert, Jon Lorsch and Michael Lauer for helpful discussion and two referees, David Currie and Christian Althaus, for useful comments on the manuscript. We are also very grateful to all those academics who responded to our request for information about their research group size.

### Funding

The authors declare there was no funding for this work.

### Competing Interests

The authors declare there are no competing interests.

## Author Contributions

- Isabelle Cook and Sam Grange performed the experiments, analyzed the data, contributed reagents/materials/analysis tools.
- Adam Eyre-Walker conceived and designed the experiments, performed the experiments, analyzed the data, contributed reagents/materials/analysis tools, wrote the paper, prepared figures and/or tables, reviewed drafts of the paper.

## Human Ethics

The following information was supplied relating to ethical approvals (i.e., approving body and any reference numbers):

It was not considered necessary to submit this study for ethical review given the nature of the project—simply requesting research group size information directly from PIs. All participants gave their written consent in the form of an email reply.

## Supplemental Information

Supplemental information for this article can be found online at http://dx.doi.org/10.7717/peerj.989#supplemental-information.

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
