# Peer review of "Research groups: How big should they be?"

_PeerJ, doi:10.7717/peerj.989_

## Round 0.1 · original submission · Major Revisions

Both reviewers had a number of specific comments. I think that all of these comments can be addressed without requiring excessive additional work.

Additional notes from PeerJ Staff:

1. Please be aware that there was some feedback on your preprint, which you may want to take account of when revising: https://peerj.com/preprints/812v1/#feedback

2. We noticed that you have spelt "principal investigator" as "principle investigator" throughout. The correct spelling is "principal".

·

Basic reporting

No comments.

Experimental design

The basic design of the study seems fine.

Validity of the findings

I have some reservations about the statistical analyses in the study. These have the potential to modify the interpretation of the results, although I doubt that the changes would be large. Please see below for details.

Additional comments

Comments re:
Cook et al. Research groups: How big should they be? Submitted to PeerJ.

I found this to be an interesting study. I have only relatively minor comments on some methodological points, and on some aspects of the interpretation.

l. 54 “doctotal” – typo
l. 79 – I agree that this error is more likely with long author lists; however, the cut-off of 20 is arbitrary.
l. 91 – “to each other” – not from each other? It is not clear what is being compare here (i.e., what “each other” refers to).
l. 94 – The p value would not be the number of differences that are in a given direction; rather, it would be the proportion. This appears to be essentially a sign test.
l. 113 – This response rate is rather low for a survey. It seems to combine non-response and response, with residency at the current institution for <5 years. It would be better to report the raw response rate (as the measure of representativeness of the sample).
l. 129 – The frequency distribution of number of papers is even more relevant than the distribution of group size. If you are going to do parametric statistics on these data, the residuals have to be homoscedastic. That seems unlikely if the parent distribution is positively skewed. In my view, it would be preferable to transform. Otherwise, the residuals from your model are likely to be heteroscedastic.
l. 136 and Fig. 2 – This figure nicely shows the problem with the skewed data (both variables). The few points with high values will have inordinately high leverage in your statistics. Transform the data. The r^2 will probably decrease. It appears that group size can be zero, which suggests that publications happen in the absence of people. The legend should specify what group size of zero represents (PI only?). The legend should also specify that these are full-time equivalents (since, otherwise, non-integral numbers of people is very curious).
l. 151-152 – I understand how you arrive at this statement, but I am not sure that I completely agree with this interpretation. Perhaps if you said that this refers to the productivity of a PI alone, relative to a PI and associates.
l. 159 – How can the number of first- or last-authored papers by a PI be greater than the average number of papers per PI estimated above (l. 150)?
L. 190 – Forward stepwise regression does not necessarily yield the best model, and there may be models that fit the data equally (not significantly differently) well. This would usually be evaluated by running models with all possible subsets of the predictor variables, and calculating AIC statistics. It may be the case that there are multiple indistinguishable models.
l. 194-195 – This almost certainly happens because there is colinearity among the numbers of different types of researchers in a lab (big groups have more of everything). Different, indistinguishable models might include a positive term for PhD students, and a negative term for some other group. Thus, I do not think that this statement is justified.
l. 212-213 – This conclusion clearly depends upon a very small number of points on the right side of this graph. I wonder if this result would still hold if the data were first transformed to eliminate the positive skew. Moreover, how much higher is the IF of a group with 10-15 people, versus a PI alone? By eye, it is something like 5.5 versus 4. This seems like an argument in favour of small.
l. 244 – Did Cohen’s analysis force the regression through the origin (i.e. not include an intercept term in the model)? That is different than fitting an intercept, and finding that it is not different than zero.
l. 274 – I am uncomfortable with the conclusion expressed this way every time I read it. This is because you have an estimate of the productivity of a PI alone, but you don’t have an estimate of the productivity of post-docs or students in the absence of PIs. Rather, you have estimates of PI-post-doc teams, and PI-student teams. So, the slopes of your regressions measure the marginal increase in productivity of a PI-student team over a PI alone. As a thought experiment, is it possible that the productivity of an isolated PI, plus the productivity of an isolated post-doc is greater than the productivity of the two together? It seems like that there is some negative colinearity in the effects of the PI and the student on productivity: viz., the PI’s independent contributions decrease when s/he has more students. Because of this, you have no way to isolate the contribution of a student or a post-dc.doc in the absence of PIs, so you cannot really say that one is X times are productive as the other. It would be more accurate to say that a PI-post-doc team is only about 30% more productive than a PI alone.

Perhaps one of the most important aspects of this study is that there is no evidence of a synergistic effect of larger research groups: there is no positive, second-degree term in your regressions. This suggests that there is no advantage in concentrating the wealth in individual labs.

David J. Currie

·

Basic reporting

No comments

Experimental design

No comments

Validity of the findings

No comments

Additional comments

This manuscript describes an interesting analysis of the relation between the size of a research group and the scientific output (publications). To this end, the authors have done a good job in collecting data from research groups in the field of biological sciences in the UK. They then used the data to assess the productivity of different members of a research group, and of research groups as a whole. While I think this study could be of broad interest, I have several critical comments that the authors should reply to before I can recommend this manuscript for publication.

1) Definitions:
The authors often talk about productivity, but do not clearly define what they mean by that. Throughout the paper, they look at the number of publications, number of citations and impact factors of journals. While those are rough surrogates for scientific output, the productivity of academic researchers could also be measured in terms of grant writing, teaching, supervision, mentoring, etc. Those other types of productivity should be discussed.

It would also help if the authors clearly define the critical variables early on in the paper. Over what period are the number of papers counted? Over what period are the number of citations counted? Is it the number of citations per paper or overall?

The manuscript implicitly assumes that the optimal research group size is one that maximizes publication output. However, the size of a research group could also be tailored such that the ratio of PhD students to postdoctoral researchers and PIs provides realistic career opportunities.

2) Technical comments:
- How did the authors calculate the explained variance from the regression coefficients? When I use r^2 as the explained variance, I obtain somewhat different values.
- Line 137: The authors argue that “the relationship between the number of papers produced and group size appears to be linear”. I don’t think that one can say that. First, there is a huge variation in the number of papers for research group of the same size. Second, the authors do not perform a statistical comparison of a series of competing models that could explain the data. What is clear is that there is a positive correlation and that this correlation can be captured with a linear increase.
- Table 1: Please indicate what statistical tests were performed and what the stars correspond to.
- Figure 2-5: It would help if the figure legends included some description of what is shown and some statistical insights, such as the slopes (including 95% confidence intervals) of the linear regressions.
- Figure 5: Wouldn’t it be interesting to look at the total number of citations per year?

3) Conclusions:
My impression is that the authors need to be more careful in interpreting their results. In the abstract, for example, they state that PIs are on average 5-times more productive than average group members. This result is not unexpected, especially if one considers that the average number of group members is 6.3. A PI will mostly be a co-author on the papers from the other group members, but not necessarily the other way around. Moreover, group members such as PhD students and postdoctoral researchers often contribute more to a paper than a PI who mainly supervises a project.

Clearly, one can question whether productivity and publication output of academic researchers at different levels in their careers can be compared at all. Looking at the number of first author papers of PIs, one would probably find that the number decreases for larger research groups. Does that mean that they are less productive? The authors need to be upfront with the various limitations of their study and discuss them in more detail. Some of the limitations should also be mentioned in the abstract in order to prevent that the results might get misinterpreted.

4) Minor comments:
- The abstract is a bit long and could be shortened substantially in order to focus on the main results and message. Also, it would help if the abstracts starts with a sentence that explains why such a study is interesting or useful.
- I could not find copies of the emails that were sent to the PIs in the supplementary data.
- Line 217: “large;ly” should be “largely”.
- Line 271-272: I find this sentence a bit exaggerated. The training of postdocs is certainly not waisted if they don’t obtain a permanent position. First, because they also provide training for PhD students, and second, because their training can be of great use in several areas outside academia.

---

## Round 0.2 · Minor Revisions

I think Reviewer 2 raises a number of valid points that should be addressed before publication. In particular, I completely agree that new results should not be introduced in the discussion and that the emails that are mentioned in the text need to be provided.

·

Basic reporting

No comments

Experimental design

No comments

Validity of the findings

No comments

Additional comments

I would like to thank the authors for responding to my previous comments. The authors have revised the manuscript substantially. Given that the analysis and conclusions have changed quite a bit, I have some new comments that should be addressed before I can recommend this manuscript for publication.

1) The authors now fit a linear model to the log-transformed data using least squares. To this end, all zeros in the data set are replaced by 1. This certainly introduces a strong bias that will likely affect their conclusions regarding diminishing returns. Personally, I find the analysis using the first model questionable, and would interpret the finding of diminishing returns with caution. The second model (fit a linear model assuming Poisson distributed data) seems much more appropriate, and I would suggest to use that model for all other analyses as well. The result from the second model also clearly indicates that the finding of diminishing returns simply arises because the PI contributes significantly more to productivity than other team members. Finally, the authors state that there is no evidence for non-linearity (line 260) because a quadratic term does not significantly improve the fit. However, there are plenty of other (non-linear) terms that might fit better. Treating the exponent as a free parameter and performing a likelihood ratio test with the (nested) linear model would provide more information.

2) Structure of the manuscript: It is confusing that you fit two models to the data, but that the models are not described at the same place (one in Materials and Methods, one in Results). I suggest you describe both models together in Materials and Methods. It is also rather unusual that you present the results of the additional analysis (regression on untransformed data) in the Discussion. I suggest you move that part to the Results.

3) I still can’t find the copies of the emails that were sent to the PIs. You also mention that “copies of the emails sent to PIs are included as supplementary information” twice in the manuscript.

4) You acknowledge David Currie and Christian Althaus by name, but you should also clarify that those were the referees of your manuscript.

5) There are plenty of typos throughout the abstract, the main text, and in the response to the reviewers’ comments. Please edit your manuscript carefully before the next submission.

---

## Round 0.3 · accepted · Accept

Thanks for addressing the remaining reviewer comments.

At the proofing stage, please make sure you proof-read everything carefully once more. While most typos have been caught in this revision, a few others remain, e.g.:
- l. 87: "those from 2011 by 2...etc" should be written as "those from 2011 by two, etc."
- l. 344: "we" should be capitalized
- In the manuscript submission system, the title of Fig. 3 is given as "Paers per group member..."